# Using Proteomics Data to Identify Personalized Treatments in Multiple Myeloma: A Machine Learning Approach

**DOI:** 10.3390/ijms242115570

**Published:** 2023-10-25

**Authors:** Angeliki Katsenou, Roisin O’Farrell, Paul Dowling, Caroline A. Heckman, Peter O’Gorman, Despina Bazou

**Affiliations:** 1Department of Electronics and Electrical Engineering, Trinity College Dublin, D02 PN40 Dublin, Ireland; ofarrero@tcd.ie; 2School of Computer Science, University of Bristol, Bristol BS1 8UB, UK; 3Department of Biology, Maynooth University, W23 F2K8 Kildare, Ireland; paul.dowling@mu.ie; 4Institute for Molecular Medicine Finland-FIMM, HiLIFE-Helsinki Institute of Life Science, iCAN Digital Precision Cancer Medicine Flagship, University of Helsinki, 00290 Helsinki, Finland; caroline.heckman@helsinki.fi; 5Department of Haematology, Mater Misericordiae University Hospital, D07 R2WY Dublin, Ireland; pogorman@mirtireland.com; 6School of Medicine, University College Dublin, D04 V1W8 Dublin, Ireland

**Keywords:** multiple myeloma, proteomics, drug sensitivity score, machine learning

## Abstract

This paper describes a machine learning (ML) decision support system to provide a list of chemotherapeutics that individual multiple myeloma (MM) patients are sensitive/resistant to, based on their proteomic profile. The methodology used in this study involved understanding the parameter space and selecting the dominant features (proteomics data), identifying patterns of proteomic profiles and their association to the recommended treatments, and defining the decision support system of personalized treatment as a classification problem. During the data analysis, we compared several ML algorithms, such as linear regression, Random Forest, and support vector machines, to classify patients as sensitive/resistant to therapeutics. A further analysis examined data-balancing techniques that emerged due to the small cohort size. The results suggest that utilizing proteomics data is a promising approach for identifying effective treatment options for patients with MM (reaching on average an accuracy of 81%). Although this pilot study was limited by the small patient cohort (39 patients), which restricted the training and validation of the explored ML solutions to identify complex associations between proteins, it holds great promise for developing personalized anti-MM treatments using ML approaches.

## 1. Introduction

Multiple myeloma (MM) is characterized by the clonal proliferation of immunoglobulin-secreting malignant plasma cells in the bone marrow. Despite high-dose chemotherapy, autologous stem cell transplants, and novel agents, the 5-year survival for MM remains below 50%, and the median survival for those with stage III disease is less than 30 months. A hallmark of the disease is the subsequent development of drug-resistant phenotypes, which may be present initially or emerge during the course of treatment and which reflect the intra-tumour and inter-patient heterogeneity of this cancer [1]. Most MM cells are initially sensitive to proteasome inhibitors (PIs), which have become the standard of care in the treatment of newly diagnosed and relapsed MM. Nevertheless, resistance (intrinsic/acquired) develops to PIs and other forms of MM therapy.

Protein biomarkers associated with sensitivity/resistance to therapy are biological measures that likely reflect the specific phenotype of the patient’s disease. For these protein biomarkers to be used to direct patient care, they must have clinical utility with very high levels of evidence, demonstrating the ability to improve clinical decision making and patient outcomes. Biomarker research in MM continues to advance in many areas and is imperative for aiding the risk stratification of patients, examining tumour evolution during progression from indolent to aggressive disease, facilitating the commencement of therapy at the most treatable stage, for the selection of therapeutic agents, and predicting the effect of therapeutic intervention with respect to sensitivity or resistance to specific agents. Next-generation proteomics are highly sensitive, less costly, and require reduced input material; thus, they will likely assist in clinical decision making in the near future. Ho et al. [2] reported that although proteomic technologies are not currently approved for clinical use in MM, there are an increasing number of studies that show great promise. Sasser et al. [3] developed a serum biomarker panel that predicts the imminent risk of multiple myeloma progression from premalignancy, whereas Bai et al. [4] identified a four-peptide panel (dihydropyrimidinase-like 2, fibrinogen α-chain, platelet factor 4, and α-fetoprotein) that predicted MM with a sensitivity and specificity of 93.55% and 92.19%, respectively, using nano-liquid chromatography, electrospray ionization, and tandem mass spectrometry (nanoLC-ESI-MS/MS). Our group has also developed a novel panel of protein biomarkers to predict responses to Bortezomib-containing induction regimens in multiple myeloma patients. Three novel biomarkers (clusterin/CLU, angiogenin/ANG, and complement 1Q/C1Q) that were predictive of Bortezomib response were identified [5]. Finally, proteomics has also lead to the identification of proteins that are altered when comparing the serum proteome from multiple myeloma patients with varying degrees of bone disease [6].

Previous studies from our group have generated proteomics data by performing mass spectrometry on MM patients’ plasma cells and grouped those data based on ex vivo drug sensitivity and resistance testing (DSRT) [7]. This The Individualized Systems Medicine approach, developed at the Finnish Institute for Molecular Medicine (FIMM), includes ex vivo chemosensitivity to 308 anti-cancer drugs, including standard-of-care and investigational drugs, with the intent to guide treatment decisions for individual cancer patients. MM patients were stratified into four distinct subgroups as follows: highly sensitive (Group 1), sensitive (Group 2), resistant (Group 3), or highly resistant (Group 4) to the panel of drugs tested [8,9,10]. Combined with the proteomic analysis of the four groups of CD138+ plasma cells, a highly significant differential proteomic signature between the four chemosensitivity profiles was identified, thus opening the way to a theranostic approach to patient treatment.

However, the study of malignant plasma cell samples from MM patients presents several challenges in terms of defining sensitive versus resistant cohorts. For example, the current International Myeloma Working Group (IMWG) criteria for assessing the response to treatment are broad and overlap between different groups. Without a clear objective delineation between the sensitive and resistance groups, the comparative proteomic statistical analysis is weakened, making it difficult to clearly identify a resistant protein phenotype. Similarly, the use of triplet combinations as a standard of care makes it difficult to identify the resistance to individual drugs [7].

Due to the significant developments in data science along with open access to medical data, machine learning (ML) has enabled the research community to investigate new medical approaches in hematology and particularly in MM, starting from diagnosis to prognosis, therapeutic intervention, and more [11,12,13,14,15,16]. For example, Povoa et al. [14] presented a multi-learning training approach that combines supervised, unsupervised, and self-supervised learning algorithms to examine the predictive value of heterogeneous treatment outcomes for MM using genes as biomarkers. Guerrero et al. [15] used ML to predict undetectable, measurable residual disease as a surrogate of prolonged survival. Ren et al. also worked on the survival prognosis [16]. They associated genes with overall survival as well as relationships between gene signature expression and common drugs used for MM.

In this paper, we present a novel methodology that explores whether proteomics could be reliably used to build ML models to infer the drug sensitivity of a patient, as outlined in Figure 1. As a first step, ML models are employed for data exploration and then for patient stratification using the patients’ proteomic profiles to identify the most accurate sensitivity groupings. This stratification confirms that proteomics can be used as features to train predictors related to drug sensitivity. We then build ML models to infer the sensitivity or resistance of each patient to a list of drugs based on their proteomic profile. These models indicate if a patient is sensitive or resistant to a drug based on their proteomic signature. Due to the small size of the patient cohort and the imbalance of the sensitive vs. resistant strata, data resampling techniques were applied, and the ML models were verified using multi-fold cross-validation.

## 2. Results

### 2.1. Dataset Exploration

The first step in building ML models is to acquire and perform exploratory data analyses. The proteomic dataset consists of 39 patients and proteomic levels for 2573 proteins for each patient. Each patient was also assigned a sensitivity group based on DSRT [7]. These groupings were Group 1 (highly sensitive), Group 2 (moderately sensitive), Group 3 (moderately resistant), and Group 4 (highly resistant). Finally, the proteins were ranked according to ANOVA-*p* values by the research team.

As part of the exploratory data analysis, the protein level data were first normalised using max–min normalisation. Then, the distributions of the top ten proteins were visualised through the histograms in Figure 2. These histograms form mixtures of distributions, which is an indicator of different patterns that are useful for patient stratification. It is evident that for most proteins, their distributions indicate that the statistical population contains two subpopulations, such as in the examples illustrated here, e.g., F13A (see Figure 2b) and MYO1F (see Figure 2e).

In addition to the histograms, a heatmap was produced to examine patients’ protein levels for many proteins simultaneously, as shown in Figure 3a. Each column in this matrix represents a patient, and each row represents a different protein from the top 100, as ranked based on the ANOVA. On the top of the heatmap, the chemosensitivity group number of each patient is included. The colourbar on the right side shows the variations in the normalised protein levels from 0 (dark blue) to 1 (bright yellow). A similar pattern in protein levels exists for patients of the same and neighbouring groups. Whereas a stark difference can be seen for patients in Group 4 compared to Group 1, it is seen that the patients in Group 4 generally have higher levels of proteins. It can also be noted that protein levels tend to either be high or low across all patients for each protein, i.e, protein 90 tends to have low levels across all patients, but protein 47 tends to be high across all patients. Based on this heatmap, it is evident that some patients’ sensitivity groups have been potentially misclassified, specifically patients 16, 17, 20, and 21. Particularly, patients 16 and 17 have been assigned to Group 3, while patients 20 and 21 have been assigned to Group 4. However, the colours showing the protein levels do not match the neighbours of these patients, indicating they have been possibly misclassified.

To verify these observations, we used a hierarchical distance-based clustering algorithm to create a dendrogram in Figure 3b. The dendrogram creates a hierarchical relational structure for the patients based on the average distances in their proteomic profiles. This is a different type of grouping than that used by the experts, which was based on chemosensitivity. Then, based on this grouping, we updated the heatmap following the ordering of the dendrogram leaves in Figure 3c. As is easily observed, the dendrogram has grouped patients 20 and 21 from Group 4 with patients from other groups. The same situation is seen for patients 16 and 17, who were grouped with patients from Groups 1 and 2. Based on these observations, an unsupervised ML approach is explored as a first step during patient stratification to relabel the patients based on their proteomic profiles.

### 2.2. Selecting Proteins as Features for ML Techniques

ML has revolutionised multiple aspects of engineering as well as modern medicine. However, it suffers from the “curse of dimentionality”, i.e., as the number of features/ dimensions grows, the number of data samples (in this case, patients) required to generalize accurately grows exponentially [17]. A rule of thumb for most machine learning approaches is that the number of features cannot be higher than the number of data samples (here, patients) [18]. If we try to do that, then the models will overfit and most probably perform poorly on a new dataset. In all the methods we tested, the accuracy for prediction was 100%, but it failed on the validation test set. Therefore, although the number of available proteins is over 2000, we cannot use all of them to build prediction models. Instead, what we try to do is understand the feature space and identify correlated features, with the aim of reducing the number of eligible features. This is a typical workflow in data mining and science [18]. Furthermore, we also rank them based on their variability, as these are more likely to lead us to a set of features that will allow us to build reliable ML models to predict a patient’s sensitivity/resistance to a drug.

A first investigation before selecting proteins for the ML techniques was to check the cross-correlation and identify potential colinearities. To this end, we computed the Pearson correlation between all proteins. We illustrate the results in the form of correlation matrices in Figure 4. High Pearson correlation values indicate potential redundancies in terms of employing those proteins as features for ML techniques. In fact, there are a variety of correlations, from very low to high, across the table. We also computed the statistical significance of those correlations through *p* values. If *p* values are lower than 0.05, then significant correlations are recorded. Around 78% of the pairs of proteins demonstrate a significant correlation. Its is therefore necessary to perform an effective feature selection for the ML technique training.

One statistical approach was to use the standard deviation of the protein levels to find the ones with the highest and lowest variability. The most variable proteins will aid in building ML models, and the least variable proteins are typically used as a sanity check. Particularly, the research group of Tierney et al. [7] used the top invariant proteins to ensure that the proteomic data collected from patients were valid. The top invariant proteins were PDCD6_HUMAN, ZW10_HUMAN, RAB2B_HUMAN, HSP7C_HUMAN, and HSP74_HUMAN. The top variant proteins will be used for the feature selection process for patient stratification. These top variant proteins were IGHA1_HUMAN, ITA2B_HUMAN, F13A_HUMAN, VINC_HUMAN, and IGLL5_HUMAN.

To confirm the findings above, various methods have been explored and tested for the ranking of all the proteins across the cohort of patients with the aim of identifying the most suitable features for the classification. Amongst those methods were Analysis of Variance (ANOVA) [19,20], Maximum Relevance Minimum Redundancy (MRMR) [21], chi square tests [22], ReliefF [23], and the Kruskar–Wallis [24] tests. All these tests resulted in non-identical feature rankings; however, there was a significant overlap in the top 50 features. We decided to use the ANOVA protein rankings for this study. The results of the ANOVA allowed for the proteins to be ranked in terms of importance when it comes to defining proteins that are most variant and, therefore, most relevant for identifying common patterns in patients. This ANOVA showed that CD9_HUMAN, ITB3_HUMAN, F13A_HUMAN, CD36_HUMAN, and NCK2_HUMAN were the top-ranked proteins in terms of statistical significance. We note that the ranking of features depends highly on the statistical method employed, and different techniques would potentially indicate a different ranking. Furthermore, from the ANOVA, it was also seen that three of the top variant proteins were included in the top 25 of the Tierney et al. [7] ranking, these being ITA2B_HUMAN, F13A_HUMAN, and CD9_HUMAN. This ranking was further verified by Pearson’s correlation coefficient [25], which produced the same ranking as the ANOVA. This ranking was used when selecting features for clustering and classification. The resulting top ten proteins based on the ANOVA *p*-values are shown in Table 1. We need to note that different ranking methods might result in a different ranking of the proteins.

### 2.3. Patient Stratification

#### 2.3.1. Patient Grouping Based on Proteomic Profiles

The original patient sensitivity groups, Group 1 (high sensitivity), Group 2 (moderate sensitivity), Group 3 (moderately resistant), and Group 4 (highly resistant), were assigned by the research team based on the patients’ responses to the panel of drugs used (DSRT platform). However, as can be seen from the heatmap and the histograms above, some patients appeared to be incorrectly assigned based on their proteomic profile patterns. To further verify this observation as well as identify patterns of proteomic profiles and determine whether those are aligned with the drug sensitivity score, we applied an unsupervised clustering method to the proteomic data. To this end, *k*-means [26,27] was used. *k*-means groups data points in *k* groups based on their similarity, expressed as the euclidean distance. In this case, *k* groups were formed based on the similarity of the protein values. To assess clustering performance, there is a set of metrics typically employed: the Silhouette (SS) [28], Davies–Bouldin (DB) [29], and Calinski–Harabasz (CH) [30] scores (see definitions and details in Section 4.2.1).

Based on the metrics presented in Table 2, the best-performing *k*-means version created two clusters using the re-visited protein ranking and ten features. These clusters could be defined as the sensitive and resistant patients. A potential increase in the number of clusters produced clustering metrics, as did increasing the number of features. Increasing the number of groupings affected the performance more than increasing the number of features. This was confirmed through visualisations of the clusters, such as the one in Figure 5. It can easily be seen that the feature ranking that resulted from our ANOVA (New ANOVA) outperformed that of Tierney et al. [7] as it enabled the formation of more distinct clusters. This is strong evidence that the new ANOVA has indicated proteins with higher variance that can help identify patterns of proteomic profiles and subsequently associations to drug sensitivity.

#### 2.3.2. Patient Grouping Based on Drug Sensitivity Scores

Next, the grouping of the patients based on their drug sensitivity score (DSS) was explored. The responses of the patients to the chemotherapeutics were ranked using the drug sensitivity scores in the same way as the proteins earlier, using ANOVA *p*-values. *k*-means clustering was then run using the top 10, 37, and 50 drugs to create groupings. Looking at the performance metrics, we noticed the same trends as with the protein clusters. The best-performing *k*-means version used ten proteins as features and created two clusters. Increasing the number of features produced worse-performing clustering metric values than increasing the number of clusters.

The best clusters created two or three chemosensitivity groups, as opposed to the four patient groups initially proposed by [7]. Due to this, clustering patients into two groups, sensitive and resistant, was investigated. The group labels produced by both the proteomics and drug data clusters were looked at alongside the original labels to divide patients into two groups. The label of the proteomics cluster was taken as the new true label as this was the best-performing cluster. Table 3 showcases the original labels, the labels produced by *k*-means clusters based on proteomics and the DSS, and our proposed labels given to patients. The majority of the DSS and proteomics groupings are aligned with the Tierney et al. [7] groupings if quantised into two: Groups 1–2 into 1 and Groups 3–4 into 2. There are a few outliers in this pattern for patients 16, 17, 21, and 22. For example, patient 21 was given by the proteomics cluster a label of 1, but their original label was 4. This outlier can also be spotted on the right side of the heatmap in Figure 3 and is one of the patients believed to be initially mislabelled. A similar situation occurs when grouping the patients into three groups. From this investigation, along with the fact that the size of the patient cohort is small, we conclude that it is more appropriate to group patients into two groups based on their drug sensitivity score: sensitive and resistant.

### 2.4. Building Prediction Models for Personalized Drug Sensitivity

The drug sensitivity score (DSS) inference problem was approached as a binary classification problem that can predict the drug responsiveness of a patient to a specific drug based on their proteomic profile. Labels were added to the dataset to create this classifier to specify if a patient was sensitive or resistant to a drug. A patient is classified as responsive and given a label of “1” for a drug if their DSS for that chemotherapeutic is 10.0 or greater. A label of “0”, indicating Group 2, given if they are unresponsive to a drug. Then, the dataset was split into three parts: the training, testing, and validation dataset. Three patients were used for the validation and the other 36 patients were roughly split into 70–30% for training and testing, respectively. During the training phase, ML model optimisation techniques, such as grid search, were employed for the model parameter optimisation.

#### 2.4.1. The Data Imbalance Issue

One of the first issues observed during the DSS label creation was that the dataset was not balanced. For example, there are 28 patients that are sensitive to Bortezomib in the testing and training sets; this is 83% of the dataset. Similarly, there are five patients sensitive to Navitoclax; this is only 15% of the dataset. This creates issues during the training of the classification models as the “minority” class cannot be properly modelled, resulting in overfitting the model to the dominant class. This would result in high accuracy, but the sensitivity and specificity would be low.

#### 2.4.2. Evaluation of the ML Models

After applying a series of different data-balancing techniques [31], we tested them on several different ML techniques, such as linear regression (LR), support vector machines (SVMs), Random Forests (RFs), neural networks, and more. We found that the most promising models were LR with One-Sided Selection (henceforth called LR+OSS), SVM with Synthetic Minority Over-sampling Technique (henceforth called SVM+SMOTE), and RF with Condensed Nearest Neighbours (henceforth called RF+CNN), as reported in Table 4. Among those combinations of techniques, the best performing method appears to be SVM+SMOTE, as it shows the highest average values of accuracy, precision, sensitivity, specificity, and the F1 metric. Looking at the confusion matrices of Bortezomib, Lenalidomide, Navitoclax, and Quisinostat for all three methods in Figure 6, we notice that in the classes with higher cardinality, the drug sensitivity is correctly predicted in most cases. For the classes of lower cardinality, it is harder to predict correctly, as not all patterns can be captured by the limited number of samples. This is captured by the lower specificity figures in Table 4.

The training data used in the models produced the ROC curves illustrated in Figure 7. From these, it is seen that the three presented ML models perform differently, and their performance varies for the different drugs. For Bortezomib, SVM+SMOTE works excellently, resulting in a AUC value equal to 1, but the other two models do not perform well, producing identical ROC curves with the same AUC value of 0.5. For Lenalidomide, the performance is significantly improved when SVM+SMOTE is employed. In this case, RF+CNN improves, but LR+OSS fails with an AUC value of 0.458. For Navitoclax, RF+CNN exhibits the best performance, but it is quite low compared to other drugs. For Quisinostat, we notice that both SVM+SMOTE and RF+CNN have a high sensitivity and specificity rate, as revealed by the ROCs with a high AUC value of 0.9, while LR+OSS does not result in high accuracy of predictions.

#### 2.4.3. Validation of the Trained ML Models

After training and testing, we validated the trained models on the three validation patients and summarise the results in Table 5. The best performing models for all three patients are the ones based on SVM+SMOTE. For patient 1 and 2, LR+OSS and RF+CNN have comparable performance; however, these models perform significantly worse for patient 3.

The results presented here indicate that the proposed approach can lead to accurate models to predict the drug sensitivity of the patients. The small size of the dataset creates constraints as not all proteomic patterns can be well represented in the model training. The resampling methods that we had to apply to compensate for the dataset imbalance further decreased the sample size (reduced to 23 samples), making it challenging to capture the complex relationship between protein levels and therapeutics. Looking at the re-sampling for RFs and LR, which both use under-sampling methods, we see that the number of samples used to train can drop to a low number for some drugs. Looking at the confusion matrices on the test dataset in Figure 6c, it is seen that this model had difficulty classifying this patient as being in the minority class for Bortezomib, which is reiterated when looking at how it classified Patient 3 in Table 5.

On the other hand, the SVM combination with SMOTE provides better results as SMOTE uses the existing minority samples to create new minority samples. Naturally, it requires a reasonable number of minority samples that capture the data. Here, SMOTE did not have enough data to accurately represent the proteomic makeup of the patients and, therefore, it was not as successful for patient 3.

## 3. Discussion

MM exhibits significant variability in both biological and clinical settings, with a notable absence of a universally applicable stratification tool for personalized treatment [7,32]. This work investigated the proteomic makeup patterns of MM patients as well as the potential to suggest personalized treatment based on those proteomic signatures. We show that proteomic data can aid in the grouping of MM patients into different chemosensitivity groups. We confirmed that using ML techniques to identify patient proteomic profile groups is more effective than using hand-picked biomarkers. The ML-based proteomic clustering resulted in meaningful groups that aligned well with the DSS. Furthermore, it proved that it is useful to use proteomic profiles to infer the DSS groups. From the vast number of proteins, it was observed that there is a strong indication of cross-correlation (and co-linearities), leading to an effective dimensionality reduction by selecting number of top-ranking proteins based on an ANOVA ranking.

One of the most significant contributions of this work is that the proposed framework has demonstrated the potential of building ML models for personalized treatment based on the proteomic profile of a patient. This is a novel finding that confirms Tierney et al.’s [7] suggestion to build predictive models that use proteomics as biomarkers. However, we note that the clinical significance of the output biomarkers should be thoroughly investigated prior to reaching the clinical setting with studies that would investigate whether these biomarkers are involved in the pathophysiology of the disease as well as whether there are drugs available for the output targets. Even more so, it should be evaluated whether a novel biomarker provides enough useful information to justify measuring it in the context of clinical care. Nevertheless, the addition of a non-specific biomarker into a biomarker panel can add to its predictive value, as reported by Lourenco et al. [33]. Furthermore, although built on a different set of biomarkers (mostly genes) and trained on a different and larger dataset, the ML-based personalized treatment framework proposed by Povoa et al. [14] did not achieve as high accuracy, precision, and sensitivity scores as our proposed model (our accuracy was, on average, 81%, while Povoa et al. [14] achieved up to 65%).

Our study provides important insights into using ML approaches for MM biomarker development; nevertheless, there are certain limitations, including the small patient cohort size and the data imbalance with regards to the different DSSs. We explored different balancing techniques that helped build better models. The best-performing ML model was the SVM, along with SMOTE as a class balancing technique. This is probably due to the way SMOTE works to create new minority samples that train the model in a more balanced way and avoid overfitting the dominant class. The limitations discussed in this work emphasise the need for open datasets, especially in rare diseases where data collection is cumbersome. Open datasets will enable clinicians and data scientists to confirm the findings around patient profile groupings and further improve the performance of DSS inference models. In addition, while proteomics-based studies undoubtedly contribute to our knowledge of the MM disease, the challenge moving forward is to not become lost in this multitude of information and be able to focus on actionable knowledge (Ho et al. [2]) to improve the outcome for our MM patients.

## 4. Materials and Methods

### 4.1. Data Collection

The proteomic dataset has been previously published by [7]. The dataset is fully anonymised and, therefore, GDPR-compliant. This dataset consists of 39 patients and proteomic levels for 2573 proteins for each patient. Moreover, for each patient, sensitivity scores are provided for 307 drugs. In this work, only Bortezomib, Lenalidomide, Navitoclax, Pomalidomide, Quisinostat, and Venetoclax were studied, as they are most frequently included in the recommended chemotherapeutic regimes.

### 4.2. Patient Groups and Stratification

For the patient grouping identification, *k*-means [26,27] was used as it groups data points based on the similarity of the proteins. *k*-means works by randomly defining *k* centroid points and categorising data points according to the centroid closest to them. These centroids are then updated iteratively to optimise the clusters. *k*-means was tested using the top 10, 37, 50, and 100 proteins as features. In addition, the number of clusters was explored. Clusters of 2, 3, 4, 5, and 6 using the different numbers of features for the new and original protein rankings were tested. Feature selection methods, such as Principal Component Analysis (PCA) [34,35], were employed to further reduce the number features (i.e., proteins).

#### 4.2.1. Metrics for Evaluation of the Strata

The formed clusters were evaluated using the Silhouette (SS) [28], Davies–Bouldin (DB) [29], and Calinski–Harabasz (CH) [30] scores. The silhouette coefficient for a single point can be defined as
(1)$$SS=\frac{b-a}{max(a,b)},$$
where *a* is the mean distance between the sample and all other points in the same class and *b* is the mean distance between a sample and other points in the next-nearest class [36]. The silhouette score is bound between −1 and 1, where the higher-scoring clusters are dense and well separated, and the lower-scoring clusters indicate incorrect clustering. If a score is around zero, this indicates an overlap in the clusters.

The Davies–Bouldin score compares the average distance between clusters with the size of the clusters themselves and is defined as:(2)$$DB=\frac{1}{k}\sum_{i=1}^{k}max\frac{s_{i}+s_{j}}{d_{ij}},$$
where *s* is the average distance between each point of a cluster, *x*, and the centroid of the cluster and $d_{ij}$ is the distance between the centroids of clusters, *i* and *j* [36]. The DB index has a lower bound of zero. Values closer to zero indicate a better separation between clusters.

The CH score expresses the ratio of the sum of between-cluster dispersion and the sum of within-cluster dispersion for all clusters, where dispersion is the sum of distances squared, as below
(3)$$CH=\frac{tr(B_{k})}{tr(W_{k})}\times \frac{n-k}{k-1},$$
where $B_{k}$ is the between-group dispersion matrix, $W_{k}$ is the within-group dispersion matrix, *n* is the size of the dataset, and *k* is the number of clusters [36].
(4)$$W_{k}=\sum_{q=1}^{k}\sum_{x\in C_{q}}\left(x-c_{q}\right){\left(x-c_{q}\right)}^{T},$$
where $C_{q}$ is a point in a cluster, *q*, and $c_{q}$ is the centre of the cluster *q*.
(5)$$B_{k}=\sum_{q=1}^{k}n_{q}\left(c_{q}-c_{E}\right){\left(c_{q}-c_{E}\right)}^{T},$$
where $c_{E}$ is the centre of the dataset and $n_{q}$ is the number of points in the cluster *q*. The CH score is not bound. High CH values indicate dense and well-separated clusters, while lower values indicate incorrect clustering.

### 4.3. Drug Sensitivity Prediction

The following types of classification models were explored for the drug sensitivity classifiers: LR, SVM with various kernels, RFs with different split parameters, ensemble trees, Gaussian processes, and forward-feed neural networks. As explained earlier, LR, SVM, and RFs exhibited better performance. An introduction to these models is included in the following paragraphs.

For all models, the dataset was split into test, training, and validation sets. The validation set contained three patients and was left untouched, while the training and testing models contained 23 and 11 patients in a 70:30 split, respectively. The dataset contained the proteomic profile of each patient and the level of sensitivity of each drug. The sensitivity level was determined based on the patient’s DSS. A patient was classified as responsive or sensitive for a drug if DSS≥10.0 for that chemotherapeutic (label of “1”). Any other DSS implied that the patient was resistant (label of “0”).

#### 4.3.1. LR

Linear regression is a basic and commonly used type of predictive analysis. It is a statistical method used to predict the value of a dependent variable based on its relationship to one or more independent variables [37]. The goal of linear regression is to find the line of best fit that minimizes the distance between predicted and actual observation values. This line of best fit is characterized by a linear regression equation that includes estimated model parameters. These parameters are estimated using a least squares approach that minimizes the sum of squared residuals between predicted and actual values. The key assumptions of linear regression models include linearity, normality, homoscedasticity, and a lack of multicollinearity [38]. Its advantages include ease of interpretation, computational efficiency, and the ability to quantify the strength of relationships. Linear regression is used for predictive modeling and forecasting across many fields, including finance, the natural sciences, and the social sciences.

For LR, the C value, the number of features selected, and the polynomial value of the features were fine-tuned. The range of the evaluated C values was {0.001, 0.01, 0.1, 1, 10}. The range of features tested was 10 to 35, with a step size of five. The polynomials explored ranged from degrees one to four. From this exploration, the model with a C value of one, 25 features, and a polynomial degree of one was chosen.

#### 4.3.2. SVMs

SVMs are a type of supervised machine learning algorithm commonly used for classification and regression tasks [39]. The goal of an SVM is to find the optimal decision boundary or hyperplane that maximizes the margin between different classes in the training data. The data points that define this hyperplane are called support vectors [40]. SVMs utilize a kernel trick to project data into higher dimensions, making them more separable. Some commonly used kernel functions include linear, polynomial, radial basis function (RBF), and sigmoid. As a discriminative classifier, SVMs draw boundaries between classes rather than modeling class probabilities generatively [41]. The key advantages of SVMs include good generalization ability, handling high-dimensional data well, and flexibility in modeling diverse data sources. Applications where SVMs excel include natural language processing, image recognition, and bioinformatics, due to their aptitude for working with small sample sizes and high-dimensional data.

Similarly to the LR, for the SVM model, the C value, the number of features, and the kernel algorithm were calibrated. The C values evaluated were {0.001, 0.01, 0.1, 1, 5, 25, 50, 75, 100}. The range of features examined was 10 to 35, with a step size of five. The kernel functions explored were linear, radial basis functions, polyonymic, and sigmoid. The radial basis function kernel with a C value of 50 using 25 features performed best for this model.

#### 4.3.3. RFs

RFs are an ensemble supervised machine learning technique used for both classification and regression tasks. They operate by constructing a multitude of decision trees during training and outputting the class that is the mode of the classes or mean prediction of the individual decision trees [42]. The training algorithm for RFs introduces randomness when building each decision tree to de-correlate with each other. This is achieved by selecting a random subset of features to consider for splits at each node and using bootstrapping or bagging to sample the training data. The key advantages of RFs include robustness to overfitting, the ability to handle many input variables without variable deletion, and effectiveness in estimating missing data and maintaining accuracy with missing data [43]. RFs have widespread applications in the fields of bioinformatics, finance, healthcare, marketing, and more.

Lastly, for RFs, the maximum number of features when looking for the best split for each tree ($max_{f}$) and the number of features used were investigated. The range of features considered was 10 to 35, with a step size of five. The $max_{f}$ range explored was from 1 to 30. Here a $max_{f}$ of five and ten features produced the best results.

### 4.4. Metrics for Evaluation of Accuracy of Prediction

The classification models were compared across several metrics: area under curve (AUC), accuracy, precision, recall, also called sensitivity in bioengineering, and F1 score [44]. The AUC calculates the area under the Receiver Operating Characteristic (ROC) curve. The ROC curve is the Recall/True Positive Rate of Equation (Equation 6) plotted against the False Positive Rate (FPR) of Equation (Equation 8) at different probability thresholds. Recall/sensitivity is the probability that a positive example is predicted to be positive.
(6)$$Recall/Sensitivity=\frac{TP}{TP+FN}$$

Sensitivity is also often seen alongside the metric specificity, which is the proportion of true negatives correctly identified.
(7)$$Specificity=\frac{TN}{FP+TN}$$

FPR is the proportion of negative values that are incorrectly labeled as positive.
(8)$$FPR=\frac{FP}{FP+TN}=1-Specificity$$

Accuracy is a measure of the probability that a prediction is correct and is given as:(9)$$Accuracy=\frac{TP+TN}{TP+FP+TN+FN}$$

Precision is the probability that a positive prediction belongs in the positive class.
(10)$$Precision=\frac{TP}{TP+FP}$$

The F1 score is the harmonic mean of precision and recall.
(11)$$F1=2\frac{Precision*Recall}{Precision+Recall}$$

When accuracy, precision, recall (sensitivity), and F1 score were calculated, each metric’s macro-average was found. The macro-average is the average of each metric calculated for each class. When these metrics are discussed, the macro-average is discussed.

### 4.5. Methodology for the Imbalanced Dataset Problem

It is seen in Section 2.4 that the imbalance in the dataset makes it difficult to create a model that can correctly identify patients in the minority sensitivity group for a drug. In order to counteract the data imbalance, several data-balancing techniques were looked at, falling into the groups of penalisation, over-sampling, under-sampling, and boosting [31]. After applying and testing these different techniques, the most promising models were LR+OSS, SVM+SMOTE, and RF+CNN. Penalisation resulted in improved results for all models, but not as good as the ones produced by the aforementioned combinations. Within the category of boosting algorithms, AdaBoost [45] was tested alongside the better-performing models; however, no improvement was reported.

#### 4.5.1. Over-Sampling

Over-sampling is a balancing technique that increases the number of samples in the minority class. This can be achieved by replicating data in that class. However, replicating data is considered unsuitable for this project due to the heterogeneity of this disease [46]. Over-sampling can also be carried out by synthetically creating data. SMOTE [47] is a technique that is often applied to biomedical data, e.g., [48,49]. SMOTE creates data by selecting the *k* nearest neighbours of a data point and interpolating between the data point and the neighbours. SMOTE, therefore, generates a data point similar to the original but not the same [46,47]. SMOTE was applied to the training data for LR, SVM, and RF to over-sample the dataset. Implementing SMOTE increased accuracy, precision, recall, and F1 in the SVM, making this the best-performing model.

#### 4.5.2. Under-Sampling

Under-sampling is a technique that reduces the number of samples taken from the majority class. This can be achieved by randomly selecting a subset of the majority of the class data [46]. However, random under-sampling like this can lead to information loss; therefore, more complex methods were used, such as Near Miss [50,51], Condensed Nearest Neighbour (CNN) [52], One-Sided Selection (OSS) [53], All KNN [54], and Instance Hardness Threshold (IHT) [53]. CNN and OSS proved to be better compared to the rest for our dataset and employed ML models.

The CNN obtains all minority samples in the training set and adds a sample from the majority class to this set *C*; all other samples are kept in set *S*. The algorithm then iterates through set *S* and classifies each sample using 1 NN. If a sample is incorrectly classified, it is added to *C*; otherwise, it is discarded. This process is reiterated until *S* is empty. The set *C* is then used as the training data [52,53]. We noted that using the CNN did not improve the performance of the SVM and LR. Nevertheless, the performance of the RF significantly improved, giving an accuracy of 0.81, a precision of 0.81, a recall of 0.78, and an F1 score of 0.79 on average.

OSS works in a similar way to the CNN but removes noisy data from the dataset instead of adding noisy data [53]. OSS improved the performance of all models when compared to the original performance but worked the best for LR.

### 4.6. Software Implementations

The majority of the algorithms and statistical methods were implemented in Python3. A few diagrams were produced in Matlab. The code and anonymised data will be made available upon request from the corresponding author.

## Figures and Tables

**Figure 1 ijms-24-15570-f001:**
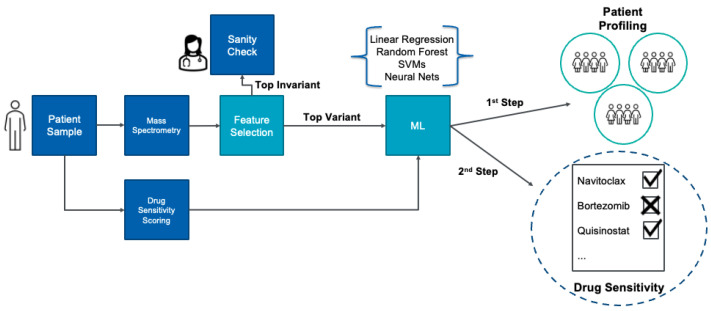
Overview of the proposed methodology that uses proteomic profiles to train ML models that can predict the potential patient strata and propose a personalized treatment plan. The patient sample is fed into mass spectrometry, which provides the responses of more than 2500 proteins. With our methodology, we select a smaller number of features to employ in our ML techniques. These ML models help us identify patient group profiles. Then, we use the selected features to build ML models to infer the drug sensitivity of a patient.

**Figure 2 ijms-24-15570-f002:**
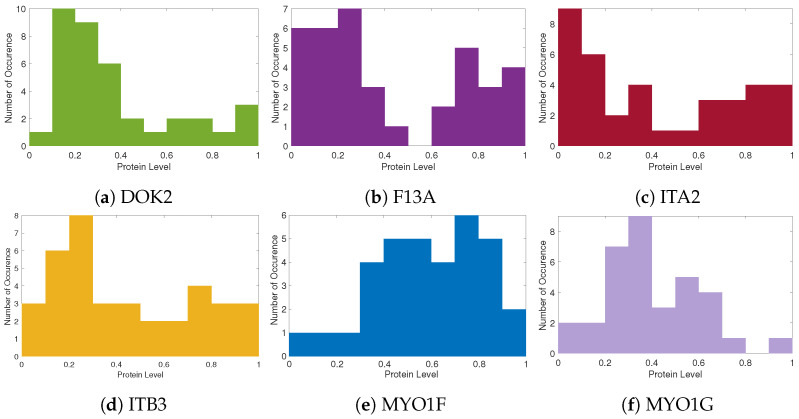
Histograms of proteomic levels across patients for named proteins. These histograms indicate mixtures of distributions, thus potential different patient proteomic profile groups.

**Figure 3 ijms-24-15570-f003:**
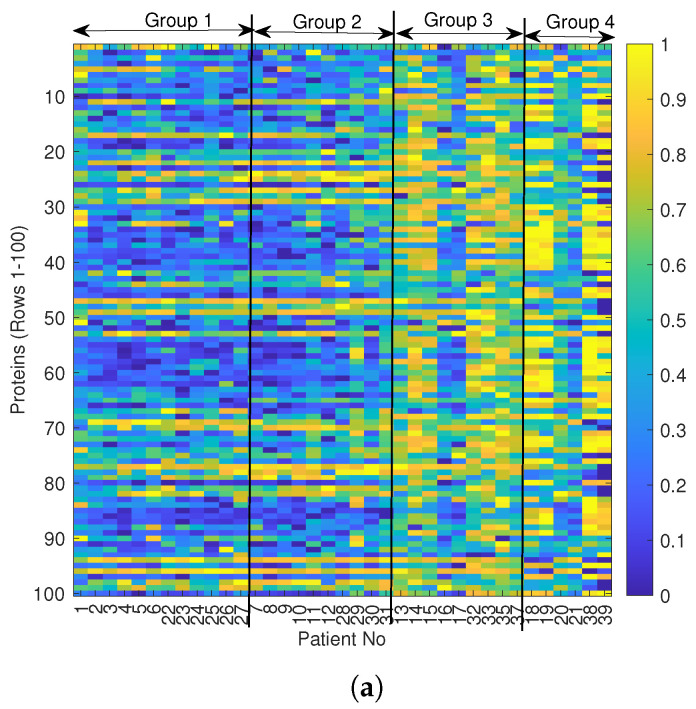
(**a**) Heatmap showing top 100 proteins for all patients: the numbering follows the group ordering based on Tierney et al. [7]; sensitive (groups 1–2) and resistant (groups 3–4) groups. The colourbar shows the range of normalised protein values. (**b**) Dendrogrambased on the average distance between proteomic profiles: the numbering of the patients on the horizontal bottom axis is not ordered, but rather follows the grouping as a result of the hierarchical clustering of the dendrogram. (**c**) Updated heatmap based on the dendrogram patient ordering.

**Figure 4 ijms-24-15570-f004:**
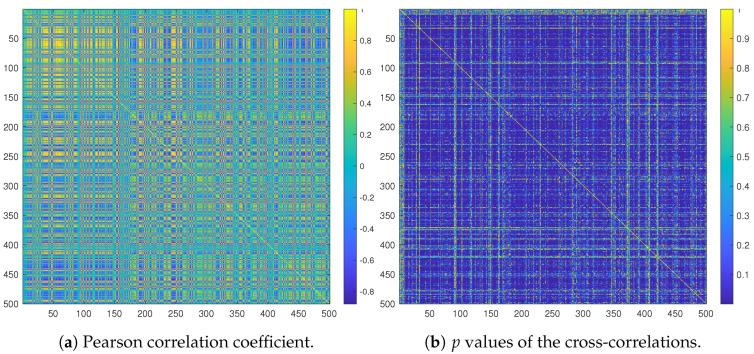
Cross-correlation and significance matrices of the first 500 proteins. (**a**) High Pearson correlation values (roughly >0.75) indicate potential redundancies in terms of the usability of the proteins as features. (**b**) *p* values lower than 0.05 (dark blue) indicate significant correlations or not.

**Figure 5 ijms-24-15570-f005:**
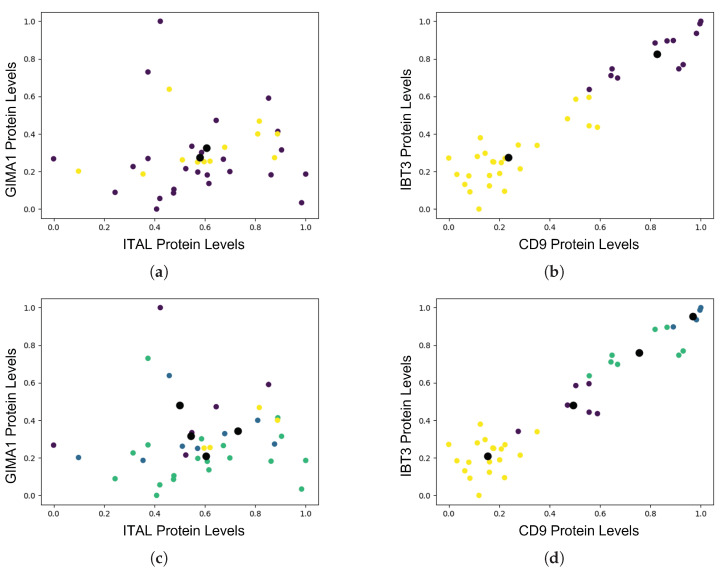
Clusters of varying numbers of patient groups with 37 features comparing feature ranking lists based on ANOVA *p*-values. The big black dots represent the centroid of the clusters. It is evident that the ranking based on our ANOVA was able to rank proteins that resulted in a clear clustering of the patients based on their proteomic profiles. (**a**) Clusters, 2; features, 37; [7] ranking. (**b**) Clusters, 2; features, 37; new ANOVA. (**c**) Clusters, 4; features, 37; [7] ranking. (**d**) Clusters, 4; Features, 37, new ANOVA.

**Figure 6 ijms-24-15570-f006:**
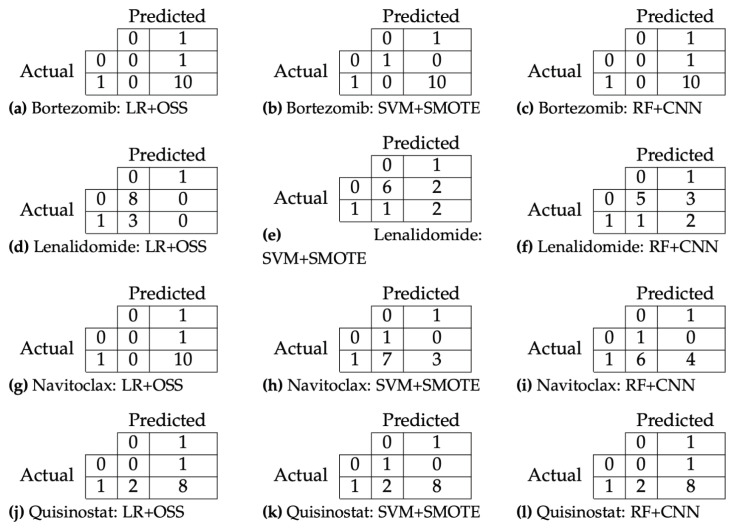
Confusion matrices of Bortezomib, Lenalidomide, Navitoclax, and Quisinostat sensitivity inference for all models. “0” indicates resistant patients, while “1” indicates sensitive patients.

**Figure 7 ijms-24-15570-f007:**
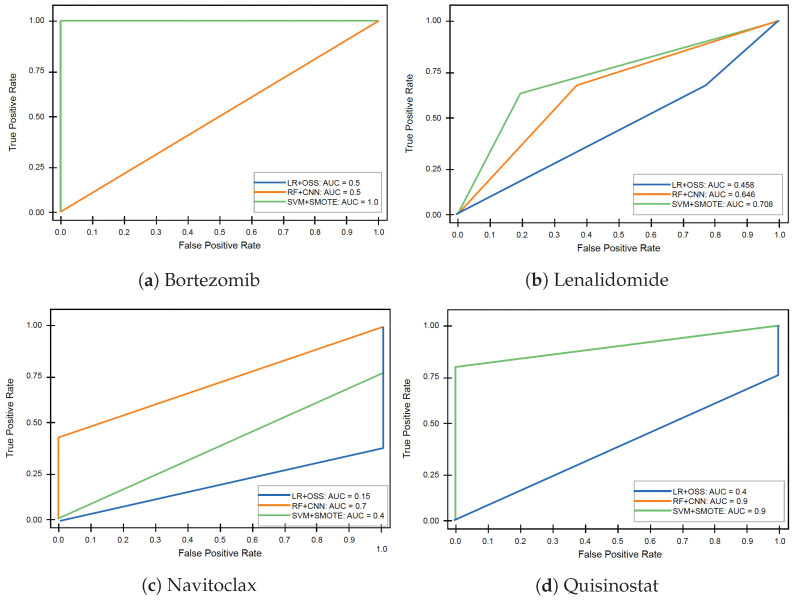
ROC curves for the trained models for Bortezomib, Lenalidomide, Navitoclax, and Quisinostat. The AUC values of all models are reported within each graph. In (**a**), LR+OSS and RF+CNN have identical performance, while in (**d**), RF+CNN and SVM+SMOTE have identical ROC curves.

**Table 1 ijms-24-15570-t001:** Table of top 10 ranked proteins based on our ANOVA *p*-values.

Top Ranked Proteins		
Rank	Protein Name	p-Value
1	CD9_HUMAN	$5.83\times 10^{-9}$
2	ITB3_HUMAN	$1.04\times 10^{-8}$
3	F13A_HUMAN	$3.91\times 10^{-8}$
4	CD36_HUMAN	$4.71\times 10^{-8}$
5	NCK2_HUMAN	$5.22\times 10^{-8}$
6	CALD1_HUMAN	$7.09\times 10^{-8}$
7	GP1BA_HUMAN	$9.38\times 10^{-8}$
8	TLN1_HUMAN	$1.64\times 10^{-7}$
9	ITA2_HUMAN	$2.67\times 10^{-7}$
10	ITA2B_HUMAN	$2.81\times 10^{-7}$

**Table 2 ijms-24-15570-t002:** Clustering performance metric results across various numbers of clusters. The best results obtained using a different number of top features are reported.

	(**a**) **New ANOVA with top 10 features**				
	**Number of Clusters**				
Metrics		2	3	4	5
	SS	0.658	0.574	0.493	0.476
	DB	0.472	0.584	0.630	0.472
	CH	126	113	109	95.6
	(**b**) **Original ANOVA with top 37 features**				
	**Number of Clusters**				
Metrics		2	3	4	5
	SS	0.329	0.207	0.167	0.183
	DB	1.22	1.49	1.65	1.40
	CH	20.8	14.3	11.7	11.05

**Table 3 ijms-24-15570-t003:** Patient group labels as derived from clustering methods applied on DSS, proteomics, and the original [7] labeling (blue denotes Group 1, yellow Group 2, green Group 3, and lilac Group 4). A few misalignments when comparing the three methods can be spotted.

ID	DSS	Proteomics	Original
1	1	1	1
2	1	1	1
3	1	1	1
4	1	1	1
5	1	1	1
6	1	1	1
7	1	1	2
8	1	1	2
9	1	1	2
10	1	1	2
11	1	1	2
12	1	1	2
13	1	2	3
14	1	2	3
15	1	2	3
16	1	1	3
17	1	1	3
18	2	2	4
19	2	2	4
20	2	2	4
21	2	1	4
22	2	1	1
23	1	1	1
24	1	1	1
25	1	1	1
26	1	1	1
27	1	1	1
28	1	1	2
29	1	1	2
30	1	1	2
31	1	2	2
32	1	2	3
33	1	2	3
35	1	2	3
37	2	2	3
38	2	2	4
39	2	2	4

**Table 4 ijms-24-15570-t004:** Average performance metric values for DSS inference with re-sampling techinques.

	Accuracy	Precision	Sensitivity	Specificity	F1
LR+OSS	0.77	0.74	0.75	0.33	0.75
SVM+SMOTE	0.81	0.79	0.8	0.61	0.79
RF+CNN	0.82	0.81	0.78	0.58	0.79

**Table 5 ijms-24-15570-t005:** Comparison of DSS classification for the three validation patients. The label “Ref” refers to the actual sensitivity of the patient, and the other labels refer to the models used to infer the sensitivity for the different drugs. The pink colour indicates the erroneous inference of the model.

	Patient 1				Patient 2				Patient 3			
	Ref	LR	SVM	RF	Ref	LR	SVM	RF	Ref	LR	SVM	RF
Bortezomib	1	1	1	1	1	1	1	1	0	1	1	1
Lenalidomide	0	1	0	1	0	1	1	0	0	1	0	1
Navitoclax	1	1	1	1	1	1	1	1	1	1	1	0
Pomalidomide	0	1	0	0	1	1	0	0	0	0	0	0
Quisinostat	1	1	1	1	1	1	1	1	0	1	1	1
Venetoclax	0	1	1	1	1	1	1	1	0	1	0	1

## Data Availability

The data presented in this study are available on request from the corresponding author. The data are not publicly available due to privacy and ethical limitations.

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
