# Peer review of "Using Proteomics Data to Identify Personalized Treatments in Multiple Myeloma: A Machine Learning Approach"

_ijms, 2023, doi:10.3390/ijms242115570_

Round 1

Reviewer 1 Report

The authors describe the use of machine learning to evaluate proteomics data in multiple myeloma patients to predict their general drug sensitivity or resistance.  The approaches are surprisingly reductionist (top 10 proteins or 10/37/50 drugs) when an outsider might view the strength of machine learning as the capability to evaluate the entire dataset.  The authors should also spend some time discussing and reporting the protein-protein correlation results with an eye to the most representative biomarker for a particular pathway or process.  The methods and results also need to be explained more for the general audience of this journal, which will include scientists with little familiarity with machine learning.  In addition, the generalization of patients as sensitive or resistant to panels of drugs is less likely to be clinically informative than the results for specific agents.  Additional explanation of the utility of the results is also needed.

Major Questions:

What happens if models are built on the proteins relevant to the drug function are compared as a control to the ML classifier?

For the 15% of patients sensitive to Navitoclax, the identification of predictive protein markers seems possible with the ML approach, but this point is not addressed in the manuscript.

Additional comparison to the results of the Tierney paper would also be helpful to show the advances from the ML approach.

Specific Comments:

Abstract: Add size of patient cohort.

Figure 3: Add dendrogram to show clustering in the heat map.

Lines 102-108:  The concerns about incorrect assignment of patient group needs more foundation, because the top 100 proteins based on ANOVA may not be the best classifier.

Lines 127-128  The immunoglobulin secreted by the myeloma cells (A heavy chain and lambda light chain) are not likely to be related to the drug sensitivity or resistance, so the top proteins should be evaluated for biological relevance prior to moving them forward in the model.  This point should be discussed as a cautionary tale in the manuscript.

Table 2 needs better annotation for the categories.  In other words, define the rows and columns with headers.

Table 3: Misalignment is really differential classification.

Table 5:  Define 0 and 1 in the confusion matrices or add a key.  

Table 6: What does the color coding mean?

Minor editing is needed for spelling, grammar, and usage.

Reviewer 2 Report

      In this research, the authors researched the potent usage of proteomics data to identify personalized treatments in multiple myeloma: a machine learning approach. In my opinion, the current version of this manuscript fits the scope of International Journal of Molecular Science and could be accepted after minor revision.

My specific comments are in detail listed below:

1.     Some minor mistakes of English usage should be checked and revised.

2.     In the discussion part, the authors could discuss the potent usage of proteomics data to identify personalized treatments in multiple myeloma by PD-L1, PD-1 or some other realted tumor immunotherapies. Some references should be added to this part including 10.1016/j.jconrel.2022.11.004.

3.     The quality of Fig. 6 was too low. Besides, the font size was small. It’s better to revise it to offer a better clear one.

4.     The clinical significance of potent usage of proteomics data to identify personalized treatments in multiple myeloma should be more clearly discussed in the discussion and introduction.
